# A Phase I Trial of Sirolimus with “7&3” Induction Chemotherapy in Patients with Newly Diagnosed Acute Myeloid Leukemia

**DOI:** 10.3390/cancers15215129

**Published:** 2023-10-25

**Authors:** Neil Palmisiano, Grace Jeschke, Lindsay Wilde, Onder Alpdogan, Matthew Carabasi, Joanne Filicko-O’Hara, Dolores Grosso, Thomas Klumpp, Ubaldo Martinez, John Wagner, Martin P. Carroll, Alexander Perl, Margaret Kasner

**Affiliations:** 1Division of Hematology and Oncology, Sidney Kimmel Medical College, Thomas Jefferson University, Philadelphia, PA 19107, USA; lindsay.wilde@jefferson.edu (L.W.); onder.alpdogan@jefferson.edu (O.A.); joanne.filicko@jefferson.edu (J.F.-O.); dolores.grosso@jefferson.edu (D.G.); thomas.klumpp@jefferson.edu (T.K.); margaret.kasner@jefferson.edu (M.K.); 2Department of Medicine, Division of Hematology and Oncology, University of Pennsylvania Perelman School of Medicine, Philadelphia, PA 19104, USAalexander.perl@pennmedicine.upenn.edu (A.P.)

**Keywords:** acute myeloid leukemia, mTORC, clinical trial

## Abstract

**Simple Summary:**

Relapsed or refractory AML remains common and difficult to treat, with no standard of care. Therefore, improving upfront therapy to prevent relapse in younger AML patients who are candidates for aggressive chemotherapy is an imperative. The PI3K/AKT/mTOR pathway helps regulate a variety of cellular processes, including protein synthesis, cell cycle progression and apoptosis. Unlike many therapeutic targets valid in only a subset of patients, this pathway demonstrates broad activation across a wide variety of subtypes of AML. Sirolimus is a highly mTORC1-selective allosteric kinase inhibitor that has been extensively studied in a wide variety of malignancies, including myeloid malignancies, and is approved as antirejection prophylaxis in solid organ transplantation. We examined levels of pS6 (a marker of activation of this pathway) at baseline and after exposure to sirolimus in patient samples, and here we report on the pharmacodynamic and clinical results of our phase 1 trial.

**Abstract:**

Chemotherapy remains a primary treatment for younger AML patients, though many relapse. Data from our group have shown that highly phosphorylated S6 in blasts may predict response to sirolimus given with chemotherapy. We report the results of a phase I study of this combination in newly diagnosed AML and the pharmacodynamic analysis of pS6 before and after treatment. Subjects received sirolimus (12 mg on day 1, 4 mg daily, days 2–10), then idarubicin and cytarabine (days 4–10). Response was assessed at hematologic recovery or by day 42 using a modified IWG criteria. Fifty-five patients received sirolimus. Toxicity was similar to published 7 + 3 data, and 53% had high-, 27% intermediate-, and 20% favorable-risk disease. Forty-four percent of the high-risk patients entered into CR/CRp. Seventy-nine percent of the intermediate-risk subjects had a CR/CRp. All favorable-risk patients had a CR by day 42; 9/11 remained alive and in remission with a median follow-up of 660 days. Additionally, 41/55 patients had adequate samples for pharmacodynamic analysis. All patients demonstrated activation of S6 prior to therapy, in contrast to 67% seen in previous studies of relapsed AML. mTORC1 inhibition was observed in 66% of patients without enrichment among patients who achieved remission. We conclude that sirolimus and 7 + 3 is a well-tolerated and safe regimen. mTORC1 appears to be activated in almost all patients at diagnosis of AML. Inhibition of mTORC1 did not differ based on response, suggesting that AML cells may have redundant signaling pathways that regulate chemosensitivity in the presence of mTORC1 inhibition.

## 1. Introduction

The treatment of acute myeloid leukemia (AML) is rapidly changing, but the standard of care for younger, fit patients remains anthracyclines and cytarabine [1]. Recently, the development of targeted agents, especially the BCL-2 inhibitor venetoclax, has broadened treatment options for patients unfit for intensive chemotherapy and with relapsed disease. Presently, with few exceptions, novel drugs have largely not altered the reliance of the upfront approach in younger adults on cytotoxic chemotherapy to induce and maintain remission in newly diagnosed AML [2]. Unfortunately, relapse still remains common, and long-term survival for these patients or those failing to enter remission remains poor [3,4,5].

The PI3K/AKT/mTOR pathway helps regulate a variety of cellular processes, including protein synthesis, cell cycle progression and apoptosis [6]. Unlike *FLT*3 or *IDH*1/2 mutations, which are potential therapeutic targets in only a subset of patients [2,7], the PI3K/AKT/mTOR pathway demonstrates broad activation across a wide variety of molecular and cytogenetic subtypes of AML in pre-clinical studies [8,9,10,11]. Interestingly, few studies have robustly determined how frequently this pathway is activated in different stages of disease (i.e., relapsed/refractory, untreated, post-transplant relapse). Largely, this lack of exploration is because of previous limits in performing analysis on the pathway in fresh cells. In AML cells, the pathway is downstream from various hematopoietic cytokine receptors, such as FLT3, c-Kit, GM-CSF and IL3, which are expressed on the cell surface of blasts. In approximately two-thirds of AML cases, recurrent mutations in genes encoding hematopoietic receptor tyrosine kinases or related proteins initiate or amplify signaling initiated by cytokine ligation to promote activation of intracellular signaling cascades [12]. Ultimately, these signaling cascades activate key downstream signal transduction targets, including PI3K and, subsequently, AKT. AKT-mediated phosphorylation of mTORC1/2 complexes allows for activation of ribosomal S6 (S6) and the eukaryotic translation initiation factor 4E-binding protein 1 (4EBP1) to regulate translation and protein synthesis [13]. Given a near-universal activation of this pathway in AML and its essential function in protein synthesis, targeting the PI3K/AKT/mTOR pathway is an attractive strategy to improve leukemia outcomes.

Sirolimus is a highly mTORC1-selective allosteric kinase inhibitor that has been extensively studied in a wide variety of malignancies, including myeloid malignancies, and is FDA-approved as an antirejection prophylaxis in solid organ transplantation [14]. Sirolimus and a related prodrug, everolimus, have single-agent activity in lymphoma, and these agents have been found to enhance chemosensitivity in vitro in human AML samples [15]. Several groups have demonstrated the feasibility of combining cytotoxic chemotherapy and mTORC1 inhibition in relapsed/refractory AML and ALL, and our group has explored whether sirolimus-induced mTOR inhibition, as measured by flow cytometric analysis of S6 ribosomal protein phosphorylation in peripheral blood blasts, predicts response to such regimens [15,16,17,18,19,20,21]. Response rates for these sirolimus-containing regimens in the relapsed/refractory leukemia setting are modest and do not appear significantly better than the traditional chemotherapy backbone regimens. As some data suggest mTOR activation is highest before disease progression [22], we postulated that newly diagnosed patients with phosphorylated S6 at baseline may benefit from the addition of sirolimus to induction chemotherapy. Herein, we report on the pharmacodynamic and clinical results of our pilot trial.

## 2. Methods

### 2.1. Trial Eligibility

We performed this prospective, single-arm, open-label, phase 1 trial, NCT01822015, with subjects enrolled at Thomas Jefferson University Hospital. The clinical protocols were approved by the Institutional Review Board at Thomas Jefferson, and all subjects signed informed consent in accordance with the Declaration of Helsinki. This trial enrolled subjects who were 18 and over with newly diagnosed, non-M3 AML who had not received prior therapy for their leukemia. An adequate Eastern Cooperative Oncology Group performance score (defined as less than 3) [23], organ function within standard limits and no active second malignancies were required for study enrollment. Subjects were prohibited from concurrently consuming medications or foods known to interact with the p450 CYP3A4 system until finished with sirolimus administration. Standard neutropenia prophylaxis antibiotics were allowed after sirolimus administration ended on day 10.

### 2.2. Treatment Plan

Subjects received a 12 mg oral sirolimus loading dose on day 1, followed by 4 mg/day on days 2–10 (see Figure 1). This dosing regimen was selected based on previous pharmacokinetic data and safety data in combination with chemotherapy developed by this group [17,18]. Standard 7 + 3 induction was given with idarubicin 12 mg/m^2^/day IV on days 4–6 along with cytarabine 100 mg/m^2^ IV via continuous infusion on days 4–10 (Figure 1) using actual body weight [24]. Response to therapy was evaluated by bone marrow biopsy and aspirate at the time of hematologic recovery or, if no count recovery, on day 42.

### 2.3. Clinical Response and Toxicities

The primary objectives of this pilot study were to determine the association between mTORC1 target inhibition during treatment and response rate to sirolimus plus idarabucin and cytarabine induction in newly diagnosed AML patients, as well as to assess the overall response of the regimen. Modified international working group (IWG) criteria were utilized for response assessment, including complete remission (CR), complete remission without platelet recovery (CRp), partial response (PR) and non-response (NR) [25]. Biomarker assessments were intended to be conducted on all patients alive at the time of response assessments.

Key secondary endpoints collected were assessment of safety, overall survival and rate of subsequent allogeneic transplantation. National Cancer Institute (NCI) Common Terminology Criteria for Adverse Events (CTCAE) Version 4.0 was used for grading of toxicities.

### 2.4. Therapeutic Drug Monitoring and Pharmacodynamics

Whole-blood sirolimus concentrations were determined using commercially available assays as performed by clinical laboratories. Sirolimus concentrations were measured from blood samples drawn 2 h after loading dose on day 1 and after dosing on day 4 (peak levels), as well as prior to day 4 trough dose. All patients tested were found to have detectable trough levels of sirolimus in the therapeutic range defined a priori as 4–12 μg/L based on previous work by this group. De-identified samples obtained at Thomas Jefferson were couriered to the University of Pennsylvania and processed within 24 h of collection. The method of PD analysis has been previously described in detail (see [16,17]). Briefly, serial tracking of mTORC1 activity was concurrently conducted alongside the monitoring of therapeutic drug levels to explore the relationship between pharmacokinetics and pharmacodynamics. The phosphorylation of S6 ribosomal protein at serine 235/6 (pS6), a downstream target of mTORC1, was quantified using flow cytometry. Initially, peripheral blood samples were obtained for measuring mTORC1 activity at the starting point (day 1 before treatment initiation) and just before administering the sirolimus dose on day 4. In cases where circulating blast cells numbered below 100 blasts per microliter, marrow aspirate was taken on both baseline and day 4 instead of peripheral blood collection. Unfractionated peripheral blood or marrow samples were immediately treated with ultrapure, methanol-free formaldehyde (final concentration 4%) to fix the cells. Subsequently, cells were permeabilized using 0.1% triton X-100 detergent at 37 degrees, washed, and preserved at −20 °C in a glycerol-based solution. Following the collection of all designated time points, samples were thawed, exposed to ice-cold methanol (90%) to enhance phospho-protein antibody signals, and subjected to flow cytometry analysis within a single cytometer session per sample.

The analysis of cytometric data was carried out utilizing FlowJo software (version 8, TreeStar). The identification of leukemic blasts was achieved through CD45 and side-scatter parameters, with at least two additional surface markers (such as CD33 and CD34) used to accurately define this cell population while excluding lymphocytes or other cell types [10]. The baseline pS6 level was established using a combination of dynamic signaling controls and staining controls. For instance, samples treated with phorbol myristate acetate, ex vivo sirolimus, and fluorescence-minus-one conditions were used as positive and negative controls, respectively. Baseline %pS6 referred to the proportion of gated blast cells exhibiting clear phosphorylation (pS6+/total gated blast events). Given the inherent heterogeneity of AML samples and the presence of subset S6 phosphorylation in unstimulated conditions, individuals whose samples exhibited >5% pS6+ events before treatment initiation were classified as having baseline mTORC1 activation. A 5% basal phosphorylation of S6 and >1000 blast events per test condition was chosen as our cutoff for baseline activation because, as previously reported, this number of blast events could adequately discriminate between signal and noise for such conditions (i.e., >10,000 cell events from an AML sample with a WBC of at least 1 K and 10% blasts, where >5% had unequivocally phosphorylated S6 at a baseline). The extent of mTORC1 inhibition was quantified by calculating the percent change in pS6-positive blasts, which was determined as 100 multiplied by [% pS6+ at baseline minus % pS6+ at day 4 trough] divided by % pS6+ at baseline.

### 2.5. Statistical Analysis

Baseline patient characteristics were reported with descriptive statistics. Scatter plots were utilized to describe baseline versus day 4 trough pS6 levels. Treatment-emergent toxicities were tabulated. The overall response rate (CR + CRp + PR) with a 95% confidence interval was calculated. The Kaplan/Meier method was utilized to describe overall survival from diagnosis date to date of last follow-up.

## 3. Results

With a median age of 62 years (range 25 to 75), fifty-five patients enrolled in this trial from 3/2013 through 10/2016. All received at least one dose of sirolimus (Table 1). Toxicity was similar to published 7 + 3 data, and prolonged aplasia without recovery was not observed (Table 2) [2,4]. Two patients (4%) died during induction from septic shock (n = 2 for unknown source) before response evaluation could be obtained. All patients included in this analysis received at least one dose of sirolimus.

Twenty-nine patients (53%) had high-risk disease, fifteen (27%) had intermediate-risk disease, and eleven (20%) patients had favorable-risk disease as defined by ELN 2017 [26]. As a whole, 35 patients (64%) entered into CR or CRp, 1 had a PR, and 19 (35%) were non-responders. The rates of response did not differ by age but did differ based on cytogenetic risk category (*p* = 0.047). Two patients (4%) died during induction. Thirteen (45%) patients with high-risk genetics had a CR or CRp, and fourteen (48%) were non-responders (Table 3).

Of the 15 subjects with intermediate-risk disease, 11 (73%) patients had a CR or CRp, 1 had a partial response, and 2 were non-responders. All favorable-risk genetics patients entered CR, and 9 of 11 were alive and in remission at the time of data analysis, with a median follow-up of 674 days. Across all risk categories, fifteen patients (27%) subsequently underwent allogeneic stem cell transplantation in CR, and eight (53% of those transplanted) were alive an average of 358 days after transplant.

Median overall survival (see Figure 2) for the group was found to be 16 months (95% CI 5.1–26.9) and was significantly different based on risk category (*p* < 0.001); 6 months for poor-risk (95% CI 2.8–9.1), 21 months for intermediate-risk (95% CI 5.8–36.1), and median NR for better-risk disease.

Forty-one of fifty-five (75%) patients had paired samples of sufficient quality for pharmacodynamic analysis (Table 4). Reasons for pharmacodynamic specimens being declared inadequate included clotted aspirate and/or inadequate cell yield for analysis. Analysis was done as previously described [16,17], with additional cell surface markers added to analyze subsets of AML cells as below. Blasts were gated using CD45 and side-scatter gating (Figure 3, left), and the percentage of leukemic blasts scoring as pS6-positive was measured (Figure 3, right). The relative decrease in the percentage of pS6-positive blasts was calculated.

In contrast to our previous results, all evaluable patients demonstrated activation of ribosomal S6 prior to therapy. Similar to prior data, S6 phosphorylation at baseline was seen in the variable percentage of blasts. In prior studies, we established a reduction of at least 40% as being associated with response to an intensive salvage regimen combining sirolimus with mitoxantrone, etoposide and intermediate-dose cytarabine (MEC). In the current study, mTORC1 was inhibited by greater than 40% in 27/41 (66%) patients overall (Table 4).

Of note, in vivo inhibition of pS6 correlates poorly with in vitro inhibition of samples taken prior to therapy and tested in vitro. Among patients with CR, 15 of 24 (63%) had >40% inhibition of S6 phosphorylation on day 4, while patients with PR or NR had a similar degree of inhibition in 11/15 patients (73%). Patients with favorable genetics showed consistent inhibition of pS6 by rapamycin (Figure 4), although one patient did not meet our threshold value of 40% inhibition. Patients with intermediate- or high-risk genetics showed variable inhibition among different patients (Figure 4) and different blast subsets (Figure 5).

## 4. Discussion

We are able to draw several important conclusions regarding the concurrent administration of sirolimus with standard induction chemotherapy in newly diagnosed AML patients. We find that the combined regimen is feasible and tolerable. Given the toxicity inherent to induction chemotherapy, our reported induction death rate of 4% is modest and comparable to what other groups have reported [27]. SAEs occurring in greater than 10% of patients largely comprised pneumonia and neutropenic sepsis, again in line with previous reports. In contrast to our previous results in a combined group of newly diagnosed high-risk and relapsed AML patients, inhibition of mTORC1 did not correlate with overall survival [28]. We hypothesize that this represents adaptive or redundant signaling in AML blasts that can modify chemosensitivity in vivo. Such adaptation has not been well appreciated in vitro and may, in fact, only be revealed by methods such as flow cytometry performed on serial samples during targeted therapies.

This study is one of the first explorations of the activation of the mTORC pathway in patients with newly diagnosed AML. Two-thirds of patients had sufficient samples for paired pharmacodynamics comparisons that were performed in real time at a second institution. Achieving higher percentages of evaluable samples was limited by a number of logistic hurdles in sample acquisition, transport and laboratory processing. Interestingly, and contrary to our expectations given our previous experience in similar samples obtained from patients with relapsed/refractory AML, all evaluable patient samples demonstrated activation of mTORC1 prior to sirolimus administration. This reinforces the idea that AML at the time of clinical presentation is characterized by activated signal transduction that includes mTOR activation, and this observation appears to be independent of karyotype. From large-scale whole-exome or whole-genome studies, it has been described that approximately 60% of patients with AML have discrete mutations in receptor tyrosine kinases or downstream targets such as RAS/MAPK signaling that are predicted to secondarily activate mTOR signal transduction. We did not study molecular subtypes by NGS. However, our findings argue that this would not be illustrative of any association between mutations and mTORC1 activation due to the universal observation of S6 phosphorylation prior to chemotherapy.

We also observed that nearly 67% of patients demonstrated significant inhibition of mTORC1 during therapy, as evidenced by a reduced amount of phosphorylated S6. We found no evidence to suggest that the presence or degree of mTORC1 inhibition correlated with an improvement in overall response or response by prognostic subcategory. Several important discoveries since the inception of this trial may explain the lack of differential response and also support continued exploration of dual mTORC inhibition in AML. Zeng et al. have described an increased in vitro and in vivo antileukemic effect of the dual mTORC inhibitor PP242 over sirolimus alone and postulate this increased activity is related to PP242′s ability to disrupt protective leukemia/stroma interactions [29]. Others have demonstrated that increased activation of AKT and mTORC2 results from the selective inhibition of mTORC1 [30,31,32]. Potentially, the increased activity of these alternative pathways may explain sirolimus’ lack of obvious effect on chemotherapy response rate or survival in this trial. mTORC1/2 kinase domain/active site inhibitors may not be associated with AKT activation; these agents are not only undergoing clinical evaluation in ALL (NCT02484430) but also may be of interest for AML therapy.

Response rates in our study were not superior to published results achieved with idarubicin and cytarabine alone, nor was the presence of degree of target inhibition predictive of response or survival [26]. Still, this does not eliminate the possibility of benefits to subsets or negate the idea that mTOR inhibition via other upstream methods of mTORC1 pathway inhibition could yield more prominent antileukemic effects. Indeed, the use of the multi-kinase/FLT3 inhibitor midostaurin is predicted to secondarily downregulate mTOR signaling [33] and has been shown to improve survival in newly diagnosed AML patients with FLT3 mutations from a placebo-controlled randomized study [2]. Our study enrolled patients prior to the widespread use of tyrosine kinase inhibitors for patients with FLT3 mutations and was underpowered to analyze FLT3 mutations as a predictor of response to or survival effects from sirolimus/idarubicin/cytarabine. Finally, even if response rates were not impacted, a substantial percentage of patients subsequently underwent transplants, and competing mortality risks from that therapy could reduce the sensitivity to detecting long-term effects of sirolimus upon survival.

## 5. Conclusions

In summary, this trial confirms the broad activation of the PI3K/mTORC1 pathway in newly diagnosed AML and supports further exploration of inhibiting mTORC1 as well as mTORC2. The feasibility of our approach to measuring mTOR inhibition provides a platform for testing other agents that primarily or secondarily inhibit mTOR activity in acute leukemia.

## Figures and Tables

**Figure 1 cancers-15-05129-f001:**
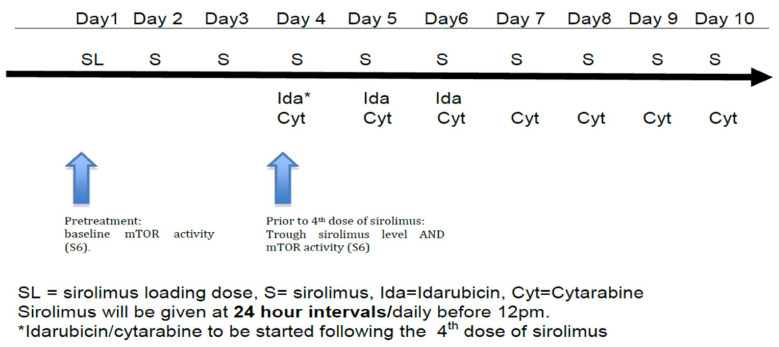
Treatment schema.

**Figure 2 cancers-15-05129-f002:**
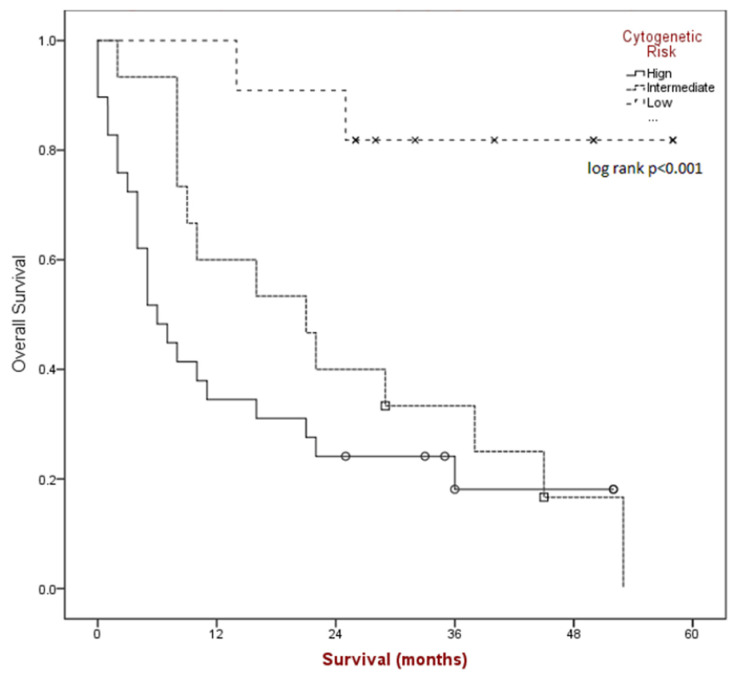
Kaplan-Meier survival estimates by cytogenetic risk category.

**Figure 3 cancers-15-05129-f003:**
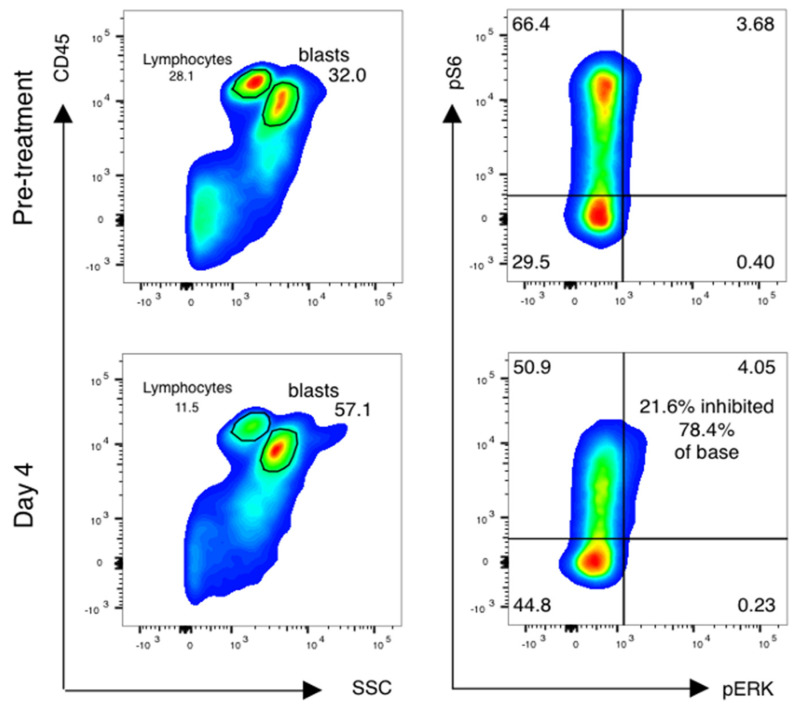
mTor inhibition in AML blasts is induced by sirolimus during therapy. Phospho-flow analysis of fixed peripheral blood collected prior to therapy and just prior to sirolimus dosing on day 4 is shown. Blasts and lymphocytes are gated based on CD45 and side-scatter properties. Cell population identity is further confirmed by staining with CD33, CD33 and CD117 and lymphocyte controls for the phosphoproteins ribosomal protein S6 and ERK are used to establish negative staining regions. The percentage of positively stained blasts on day 4 is compared to baseline to identify a treatment-induced inhibition percentage in phosphorylated S6.

**Figure 4 cancers-15-05129-f004:**
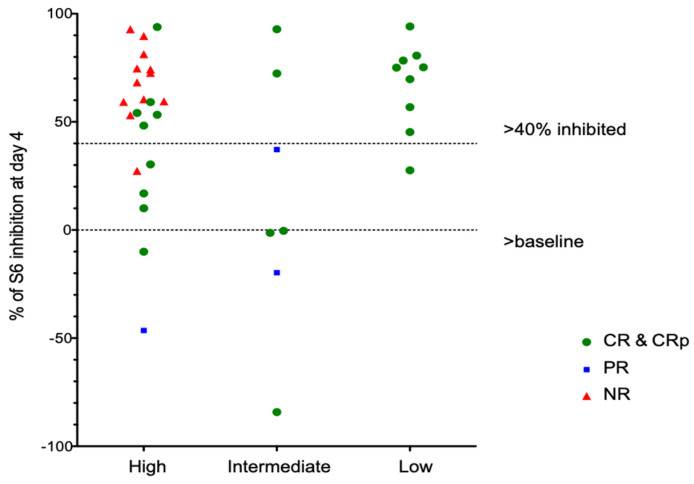
pS6 Analays stratified by AML cytogenetic risk group.

**Figure 5 cancers-15-05129-f005:**
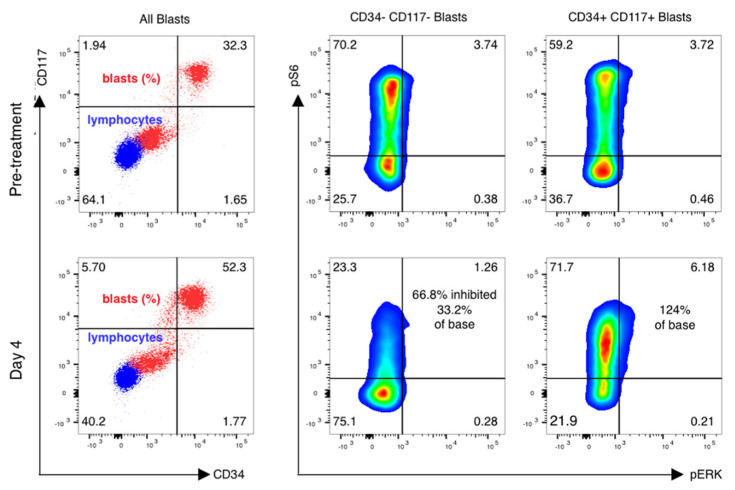
pS6 activation and inhibition vary in distinct leukemic subsets of cells. mTOR inhibition is variable in blast subsets. Phospho-flow analysis of fixed peripheral blood collected prior to therapy and just prior to sirolimus dosing on day 4 is shown in gated blasts identified by CD45 and side-scatter properties. CD45 dim, low-scatter blasts were confirmed to express CD33 but have variable CD34 and CD117 expression, with basal activation of mTOR seen in both CD34+/CD117+ blasts as well as CD34−/CD117− blasts. Sirolimus induced a marked reduction in ribosomal protein S6 phosphorylation in CD117− /CD34− gated blasts only, with activation of S6 and ERK phosphoprotein increasing on day 4 of therapy in CD34+/CD1117+ blasts.

**Table 1 cancers-15-05129-t001:** Baseline patient characteristics.

AML	=55
Age—median	26 (21–75%)
Gender	
Male	34 (62%)
Female	21 (38%)
Disease status at diagnosis	
De novo	50 (91%)
sAML/tAML	5 (9%)
Risk Classification	
Favorable	11 (20%)
Intermediate	15 (27%)
Poor	29 (53%)

**Table 2 cancers-15-05129-t002:** Common (>10%) treatment emergent adverse events.

	All Grades		Grade 3/4	
	N	%	N	%
Diarrhea	32	58.2	0	0
Febrile Neutropenia	28	50.9	21	38.2
Nausea	28	50.9	0	0
Anorexia	24	43.6	1	1.8
Mucositis	17	30.9	1	1.8
Rash	16	29.1	1	1.8
Vomiting	16	29.1	0	0
Hyperglycemia	15	27.3	0	0
Hyperbilirubinemia	13	23.6	3	5.5
Hypokalemia	13	23.6	2	3.6
Elevated ALT	12	21.8	1	1.8
Fatigue	12	21.8	0	0
Coagulopathy	11	20	0	0
Hypophosphatemia	10	18.2	6	10.9
Acute Kidney Injury	9	16.4	1	1.8
Constipation	9	16.4	1	1.8
Hypertension	9	16.4	5	9.1
Epistaxis	8	14.5	0	0
Elevated Alkaline Phosphatase	7	12.7	0	0
Hypocalcemia	7	12.7	1	1.8
Hyponatremia	7	12.7	1	1.8
Pneumonia	7	12.7	6	10.9
Abdominal pain	6	10.9	1	1.8
Elevated AST	6	10.9	1	1.8
Hematuria	6	10.9	0	0
Sore throat	6	10.9	0	0

**Table 3 cancers-15-05129-t003:** Response by Cytogenetic Risk Category.

	High Risk (n = 29)	Intermediate Risk (n = 15)	Favorable Risk (n = 11)	Total (n = 55)
CR/CRp	13	11	11	35 (64%)
PR	0	2	0	1 (2%)
Non-Response	14	2	0	17 (30%)
Death in Aplasia	2	0	0	2 (4%)

**Table 4 cancers-15-05129-t004:** Decrease in pS6 Does Not Correlate With Clinical Response in De Novo AML Patients Treated with Sirolimus Prior to Induction Chemotherapy.

Phospho-Flow Data by Risk Category									
	High Risk		Intermediate Risk		Favorable Risk		All-Comers		
n total	29		15		11		55		
n evaluable	25	86%	7	47%	9	82%	41	75%	of total
n active	25	100%	7	100%	9	100%	41	100%	of evaluable
n inhibited	17	68%	2	29%	8	89%	27	66%	of evaluable
CR[p], n	13	45%	11	73%	11	100%	35	64%	of total
CR, evaluable	10	77%	5	45%	9	82%	24	69%	of subset total
CR, inhibited	5	50%	2	40%	8	89%	15	63%	of evaluable
PR, n	1	3%	1	7%	0		2	4%	of total
PR, evaluable	1	100%	0	0%			1	50%	of subset total
PR, inhibited	0	0%					0	0%	of evaluable
NR, n	13	45%	3	20%	0		16	29%	of total
NR, evaluable	12	92%	2	67%			14	88%	of subset total
NR, inhibited	11	92%	0	0%			11	79%	of evaluable
died in aplasia	2	7%	0		0		2	4%	of total

## Data Availability

The data presented in this study are available on request from the corresponding author.

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
