# Peer review of "A Phase I Trial of Sirolimus with “7&3” Induction Chemotherapy in Patients with Newly Diagnosed Acute Myeloid Leukemia"

_cancers, 2023, doi:10.3390/cancers15215129_

Round 1

Reviewer 1 Report (Previous Reviewer 4)

The authors have adequately addressed the concerns that had been raised.

Author Response

Thank you for your kind re-review. 

Reviewer 2 Report (Previous Reviewer 1)

In this manuscript entitled “A Phase I Trial of Sirolimus with “7&3” Induction Chemotherapy in Patients with Newly Diagnosed Acute Myeloid Leukemia”, Palmisiano et al., reported the results of a phase I clinical trial of using mTor 1 inhibitor Sirolimus in treatment of newly diagnosed AML patients in combination with standard chemotherapy.  Although addition of Sirolimus failed to improve the clinical response of patients, the conclusion of this study is solid and worth to report.  Many previous studies demonstrated that most tumor cells quickly acquired resistance to mTor1 inhibitor treatment due to a feedback activation of compensatory signaling.  However, the inhibition of pS6 was examined only at day 4 of the treatment,  please discuss whether failure of inhibition of pS6 in AML blasts in most cases are due to the insufficiency of Sirolimus dose or due to the feedback activation mechanism. 

Major concerns:

1.      Figure 3. “Lymphocyte controls for p-S6 and p-ERK are used to establish negative staining regions (data not shown).” Please show this data so that the audiences can see how the experiments were performed. 

2.     Fig 4. Of note, in vivo inhibition of pS6 correlates poorly with in vitro inhibition of samples taken prior to therapy and tested in vitro (data not shown).  Please also show the in vitro data to demonstrate how the results from in vitro study failed in predicting the in vivo response.

3.     In AML blasts, mTor1 signaling activation can be stimulated either mutations in receptor tyrosine kinases, or downstream targets such as RAS/MAPK signaling, or cytokine signaling stimulation.  These different types of mTor1 signaling activation might have different biological functions.  It will be better if the authors can provide the information of receptor tyrosine kinase mutations of the AML cases and verify their conclusion: there is not any association between mutations and mTORC1 activation.  Whither receptor tyrosine kinase mutations are associated with therapeutic response need to be verified. 

Some typo

Author Response

We thank the reviewer for their comments.   Our responses to number questions:

  1. We have previously published this data, and do not feel that it substantially improve the conclusions of this report. Perl, Clin Can Res. 2012.the
  2. 100nM sirolimus we used for ex vivo spike in experiments to estimate the maximal S6 inhibition of mTORC1 by rapamycin exceeds safe human concentrations by ten-fold. The lack of correlation with clinical results observed after 3 days of oral sirolimus at therapeutic levels for transplant immunosuppression is not surprising.  We respectfully disagree that showing this data helps in interpreting or the discussion of the ultimate findings of this study.
  3. We agree that having more robust mutational data would enhance the analysis and perhaps identify subsets of responders vs non-responders.  Patients were initially recruited starting in 2013 before our institution broadly performed NGS sequencing.  We do not have this data to analyze and furthermore, given the relatively small number of subjects on the study compared to largely number of potential mutations, we doubt that performing these studies now would result in statistically meaningful findings. 

Reviewer 3 Report (Previous Reviewer 2)

The authors have mostly addressed my comments from the first round of reviews. 

  The years for conduct of the study should be included. This would be helpful for context given other studies using sirolimus with chemotherapy for AML. Although addressed in the "response to review" I still don't see these dates included. 

Author Response

Thank you for your kind re-review.  Due to an editing error the updated years of accrual was inadvertently left out.  It has been added to this version of the manuscript (ln 176).

Reviewer 4 Report (Previous Reviewer 3)

The paper is well-written and clear.  I did not see a mention of when the study was done ("prior to the widespread use TKI for FLT-3"). 

Author Response

Thank you for your kind re-review.  Due to our editing error,  the updated years of accrual was inadvertently left out.  It has been added to this version of the manuscript (ln 176).

Round 2

Reviewer 2 Report (Previous Reviewer 1)

All my concerns were addressed.

This manuscript is a resubmission of an earlier submission. The following is a list of the peer review reports and author responses from that submission.

Round 1

Reviewer 1 Report

In this manuscript, Dr. Palmisiano et al., summarized a phase I clinical trial of addition Sirolimus, a mTor inhibitor in “7&3” induction chemotherapy in treatment of newly diagnosed acute myeloid leukemia.  Total 55 cases were studied.  Although this new regimen does not improve the treatment effect and overall survival of patients,  the conclusion is important and interesting.  The clinical data is well summarized and manuscript is well-written, despite the weakens in laboratory studies.  Several concerns need to be addressed before accepted for publishing. 

1.        Please verify the numbers in table 3 to make corrections. 

2.       Figure legends are required for all Figures in order to let the readers known how the experiments were conducted.   

3.       In page 3 lines 136, “Toxicity was similar to published 7+3 data”.  Please provide references or additional data to support this conclusion.

4.       In page 4 lines 146-148, “ Two patients (4%) died during induction. Thirteen (44%) patients with high-risk genetics had a CR or CRp, 14 (43%) were non-responders, and 2 died in  induction (Table 3)”.  Please verify the numbers and %.  If 13 patients is 44%, 14 patients could not be 43%.   “Two patients died” has been described for several times in the manuscript and redundant. 

5.       In page 5 lines 151-152,  “Of the 15 subjects with intermediate risk disease, 11 (79%) patients had a CR or CRp, had a partial response, and 2 were non-responders.” The numbers and % need to be verified. 

6.       In page 5 lines 154-156, “Fifteen patients (27%) subsequently underwent allogeneic stem cell transplantation in CR, and 8 (53% of those transplanted) are alive an average of 358 days after transplant.”  I believe this discusses the total 55 patients, please clarify.

7.       Figure 3.  pS6 and pErk were examined in blasts only.  It might be better to include pS6 and pErk in lymphocytes as controls.  And also It is better to combine Figures 3 and 4 as one Figure.

8.       Figure 5. The % of CD34+CD117+ blasts is increased after Sirolimus treatment which seems associated with increased % of pS6 cells.  This is very important information, please verify whether this is the case for all samples and is related to therapeutic response.

9.       In many early studies, feedback activation of upstream signaling have been defined as a major reason for acquired resistance to Sirolimus.  It is better to provide some additional experimental information whether AKT-mTor2 signaling is activated after Sirolimus treatment.

Author Response

Thank you for your kind review.

  1. Please verify the numbers in table 3 to make corrections.  
    • Verified and updated.
  2. Figure legends are required for all Figures in order to let the readers known how the experiments were conducted.  
    • Completed and updated in manuscript.
  3. In page 3 lines 136, “Toxicity was similar to published 7+3 data”. Please provide references or additional data to support this conclusion.
    • Reference added and discussed in manuscript. 
  4. In page 4 lines 146-148, “ Two patients (4%) died during induction. Thirteen (44%) patients with high-risk genetics had a CR or CRp, 14 (43%) were non-responders, and 2 died in induction (Table 3)”.  Please verify the numbers and %.  If 13 patients is 44%, 14 patients could not be 43%.   “Two patients died” has been described for several times in the manuscript and redundant. 
    • Corrected percentages and redundancy deleted. 
  5. In page 5 lines 151-152, “Of the 15 subjects with intermediate risk disease, 11 (79%) patients had a CR or CRp, had a partial response, and 2 were non-responders.” The numbers and % need to be verified. 
    • Numbers/% verified and edited. 
  6. In page 5 lines 154-156, “Fifteen patients (27%) subsequently underwent allogeneic stem cell transplantation in CR, and 8 (53% of those transplanted) are alive an average of 358 days after transplant.” I believe this discusses the total 55 patients, please clarify.
    • Edited for clarity in the body of manuscript. 
  7. Figure 3. pS6 and pErk were examined in blasts only.  It might be better to include pS6 and pErk in lymphocytes as controls.  And also It is better to combine Figures 3 and 4 as one Figure.
    • We agree that negative controls allow for more accurate estimation of phopho-signal in tumor cells. However, in essentially all samples, lymphocytes have minimal to no basal activation of either pS6 or pERK and therefore we have used them as negative controls for each stain. The edge of the lymphocyte population used to define a boundary of the negative population's staining. We respectfully disagree with the reviewer that overlaying the negative population on the blasts would add substantively to the color enhanced plots on the figure as they would be rendered monochromatic, and it can be harder to see trends in the blast staining if cell events are limited. We have however shown this gating strategy in prior publications with phopho-S6 (Perl AE, et al Clin Cancer Res. 2012 Mar 15;18(6):1716-25) and similar methods were published using phosphorylated ERK in both AML blasts and lymphocytes by David Hedley's lab (Tong FK, et al. Cytometry B Clin Cytom. 2006 May;70(3):107-14). We have clarified this point in the methods to allow the reader to review our rationale behind the gating strategy.
  8. Figure 5. The % of CD34+CD117+ blasts is increased after Sirolimus treatment which seems associated with increased % of pS6 cells. This is very important information, please verify whether this is the case for all samples and is related to therapeutic response.
    • Changes in observed CD34+/117+ frequency after 3 doses of sirolimus were not present at day 4 for all samples, just this one. We do not consider this to be predictive of response. 

  9. In many early studies, feedback activation of upstream signaling have been defined as a major reason for acquired resistance to Sirolimus. It is better to provide some additional experimental information whether AKT-mTor2 signaling is activated after Sirolimus treatment.
    • We thank the reviewer for this comment, which indeed is particularly relevant for long-term single-agent mTOR inhibitor therapy, such as everolimus treatment of renal cell carcinoma, neuroendocrine, or other solid tumors. We would note that, beyond the different tumor types, mechanisms of resistance here to sirolimus plus cytarabine and idarubicin may differ due to the differences of study design, particularly our use of a combination chemotherapy regimen to which sirolimus is potentiating a cytotoxic effect.

      To directly answer the reviewer's question, we did not examine our samples to determine if mTORC2 complex (and hence AKT) was activated via feedback in leukemic blasts following response to sirolimus for several reasons. First, the duration of sirolimus monotherapy examined by phospho-flow was short-term and limited to 72h of sirolimus exposure on this trial. Feedback activation of TORC is generally thought to require longer-term exposure to activate AKT. Second, and perhaps more importantly, basal levels of pAKT are extremely low in primary AML blasts by phospho-flow and flow-validated phospho-PRAS40 antibodies to date are not available. Using our methods, we could not reliably detect basal pAKT activation in samples above the level of lymphocytes without the use of cytokine stimulation. This was true even in samples where constitutive pAKT could be demonstrated by western blot.  Accordingly, we do not think there is sufficient signal to noise ratio by phospho-flow to evaluate mTORC2 activation state accurately using our methods.

Reviewer 2 Report

Palmisiano et al report results of a Phase 1 study of sirolimus with 7+3 induction for newly diagnosed patients with AML. The study enrolled 55 patients. The investigators measured pS6 by flow cytometry at baseline and with treatment. This study follows similar studies conducted by this group for patients with r/r AML and high risk AML. In this study the level of pS6 does not seem to correlate with response raising the question of whether this is a helpful biomarker. Overall, this is a well conducted study and certainly adds to the literature of similar regimens.

1.     The years for conduct of the study should be included. This would be helpful for context given other studies using sirolimus with chemotherapy for AML.

2.     This group of investigators conducted two studies of sirolimus+MEC for patients with AML with r/r leukemia or newly diagnosed high risk AML, published in 2018. The authors should reference this earlier study in the current manuscript. There is some discussion about similarities/differences in the pS6 response- how does this contribute to trial analysis? Is measurement of pS6 predictive at all?

3.     This is a Phase 1 trial thus primary objective should be safety. Please clarify section 3.3 with the actual primary objectives of the study. It seems unusual to enroll newly diagnosed patients with AML on a Phase 1 trial—please clarify. The trial enrollment of 55 patients also seems to be quite high for a Phase 1 trial, especially where there’s no dose escalation. The Phase 2 study conducted by this group (published in 2018) was smaller.

4.     How does the response to this combination compare to 7+3 without sirolimus? This should be commented on.

5.     The flow plots show pERK flow but this is not discussed. Did pERK correlate with response at all?

6.     Line 234 is unclear. Should it be “doesn’t eliminate”?

Line 196/197: typos in adaptive

Author Response

Thank you very much for your kind review of our work.  Our responses are below. 

  1. The years for conduct of the study should be included. This would be helpful for context given other studies using sirolimus with chemotherapy for AML.
    • This information has been added to the manuscript.

  1. This group of investigators conducted two studies of sirolimus+MEC for patients with AML with r/r leukemia or newly diagnosed high risk AML, published in 2018. The authors should reference this earlier study in the current manuscript. There is some discussion about similarities/differences in the pS6 response- how does this contribute to trial analysis? Is measurement of pS6 predictive at all?
    • This study has been added to references and is discussed in section 4.

  1. This is a Phase 1 trial thus primary objective should be safety. Please clarify section 3.3 with the actual primary objectives of the study. It seems unusual to enroll newly diagnosed patients with AML on a Phase 1 trial—please clarify. The trial enrollment of 55 patients also seems to be quite high for a Phase 1 trial, especially where there’s no dose escalation. The Phase 2 study conducted by this group (published in 2018) was smaller.
    1. While typically phase I trials are aimed at assessing safety as their primary objective, this trial was designed to obtain samples for pharmcodynamic sampling.  We continuous monitored for safety and this was a secondary enpoint.  The dose selected 4mg had previously been optimized in previous studies by the group and this language has been added to the manuscript.

  1. How does the response to this combination compare to 7+3 without sirolimus? This should be commented on.

  1. The flow plots show pERK flow but this is not discussed. Did pERK correlate with response at all?
    1. Discussion of pERK has been added to fig3 and fig5 legend. 
  1. Line 234 is unclear. Should it be “doesn’t eliminate”?
    • Edited for clarity

Reviewer 3 Report

The authors are to be applauded for continuing to pursue the inhibition of mTORC1 pathway in the treatment of AML.  This study follows earlier work from some of the authors published over a decade ago at the time focusing on relapsed/refractory AML.  Demonstrating an improvement in response/outcome in untreated AML is a more challenging task.  That noted, the failure of this study to show an impact on 7+3 with Sirolimus is not surprising since challenging 7+3 has not met with success except in the case of CBF AML, FLT-3 mutated AML, and IDH mutated AML.

It would be useful to know the period of time over which this study was performed since it seems that it was started long ago.

The concluding paragraph states that is this study confirms that broad activation of mTORC1 activation is found in R/R AML but this study was in newly diagnosed AML so this is not accurate. 

Author Response

Thank you for your kind review of our manuscript.  Below are the responses to your helpful comments. 

It would be useful to know the period of time over which this study was performed since it seems that it was started long ago.

  • Added to manuscript

The concluding paragraph states that is this study confirms that broad activation of mTORC1 activation is found in R/R AML but this study was in newly diagnosed AML so this is not accurate. 

  • Edited discussion for clarity

Reviewer 4 Report

In this manuscript, the authors tested the effects of the addition of the mTORC1-selective allosteric kinase inhibitor sirolimus to standard induction chemotherapy for AML would enhance the efficacy of this regimen.  They found that the regimen was well tolerated, but that inhibition of mTORC1 did not correlate with response.

This study tested an interesting and under-explored hypothesis related to AML biology and treatment.  The pharmacodynamic analyses were well-designed and executed, and the knowledge generated will be useful in subsequent analyses of AML.  In addition, the following points should be considered:

1.     It would be helpful to indicate when patients were enrolled on this trial. The 2017 ELN risk stratification by genetics is utilized for cytogenetic risk categorization in this study. However, information regarding gene mutations, which is recommend in the 2017 ELN guidelines, is not included. Analysis incorporating molecular testing could augment the significance of this data, especially since redundant signaling pathways are postulated in the text. The molecular profiling of responders compared to non-responders should also be provided, if available. 

2.     How was the sirolimus dosing schema determined? Providing the rationale for this regimen would be helpful to investigators in this area.

3.     Were sirolimus levels monitored after day 4?

4.     It would be helpful to indicate why a threshold of >5% pS6+ events was chosen, and whether there was in vitro data to support this decision.

5.     It would be useful to expand upon why in vivo inhibition of pS6 correlated poorly with in vitro inhibition.

6.     Including additional information regarding response rates of the cited sirolimus-containing regimens in the relapsed/refractory setting would be helpful. 

7.     It would be useful to include the etiology of sepsis in the two patients who experienced death in aplasia.

8.     The final paragraph should be clearer that this study applies to newly diagnosed patients.

9.     Based on the experience generated from this study, it would be helpful if the authors could be more specific in delineating what the next questions might be in this field.

Author Response

Thank you for your thoughtful review of this manuscript.  Below are our responses to your helpful comments/questions.

  1. It would be helpful to indicate when patients were enrolled on this trial. The 2017 ELN risk stratification by genetics is utilized for cytogenetic risk categorization in this study. However, information regarding gene mutations, which is recommend in the 2017 ELN guidelines, is not included. Analysis incorporating molecular testing could augment the significance of this data, especially since redundant signaling pathways are postulated in the text. The molecular profiling of responders compared to non-responders should also be provided, if available. 
    • Agree with the reviewers comments, however, aside from the molecular markers in ELN 2017, broad NGS panels were not reliable completed. Given small numbers of any particular mutations included in ELN 2017 criteria, correlation was not performed.
  2. How was the sirolimus dosing schema determined? Providing the rationale for this regimen would be helpful to investigators in this area.
    • Previously studies in the relapsed/refractory setting performed by our group were used as the basis for this schema. The manuscript has been updated and clarified to reflect this.
  3. Were sirolimus levels monitored after day 4?
    • No further monitoring after determination of trough sirolimus level was conducted.
  4. It would be helpful to indicate why a threshold of >5% pS6+ events was chosen, and whether there was in vitro data to support this decision
  5. It would be useful to expand upon why in vivo inhibition of pS6 correlated poorly with in vitro inhibition.
    • Please note that the follow response is a combined to answer questions 5 and 6 and the manuscript has been updated accordingly: 

      The phospho-flow assay using pS6 for a readout was linear in cell lines over a large range. Side by side testing in cell lines shows that the flow readout has a much greater dynamic range than Western blot. However, it was not possible with sirolimus alone to get down to low pS6 levels in AML cell lines and correlate this with western blot. This is because treating cell lines with this drug leads to cells with suppressed, but residual S6 phosphorylation by flow, even if WB shows no residual phospho-signal. AML cell lines generally have very high basal levels of pS6 by flow and several signaling pathways appear to regulate S6 phosphorylation. For example, we could not drive pS6 to zero in AML cell lines by flow through inhibiting mTORC1 alone with rapamycin alone, but easily could show this by inhibiting both mTORC1 and also MAPK signaling (i.e. exposing cell lines to a combination of rapamycin and the MEK inhibitor U0126).

      In primary AML samples, basal pS6 levels are quite variable by flow and often very low in comparison to cell lines. At very low basal pS6 levels with limiting numbers of cell events, it can potentially can be difficult to tell if the basal pS6 signal reflects an artifact of whole blood processing, fixation, permeabilizing, etc. Therefore, we wanted confidence that a baseline sample was evaluable for any therapy-induced reductions in S6 phosphorylation.

      As we published previously, while developing this assay, we chose 5% basal phosphorylation of S6 and >1000 blast events per test condition as our bare minimum. This was because we felt this number of blast events adequately could discriminate between signal and noise for such conditions (ie >10,000 cell events from an AML sample with a WBC of at least 1K and 10% blasts where >5% of had unequivocally phosphorylated S6 at a baseline). By this, we mean at least 50 blast events with unequivocally phosphorylated S6 were considered a pre-requisite for an evaluable sample such that a change from this baseline would be confidently read as real and not noise.  Running multiple baseline samples sequentially showed that, even at these low levels, basal phosphorylation changes from tube to tube were minimal.

  6. Including additional information regarding response rates of the cited sirolimus-containing regimens in the relapsed/refractory setting would be helpful. 
    • This is now addressed/referenced in introduction.
  7. It would be useful to include the etiology of sepsis in the two patients who experienced death in aplasia.
    • Added to manuscript.  In both cases an etiologic agent of sepsis could not be determined.
  8. The final paragraph should be clearer that this study applies to newly diagnosed patients.
    • Clarified
  9. Based on the experience generated from this study, it would be helpful if the authors could be more specific in delineating what the next questions might be in this field.  Discussion updated.